

# Activation of autophagy is required for clearance of mitochondrial ROS in patients with asthenozoospermia

Xiaona Wang[1],*, Shiyuan Huang[2],*, Yu Zhao[1], Hua Chen[1], Linzhi Yan[1], Hongshan Ge[3] and Xinmei Wu[4]

[1] Department of Obstetrics and Gynecology, The Second Affiliated Hospital and Yuying Children's Hospital of Wenzhou Medical University, Wenzhou, China
[2] Department of Neurological Rehabilitation, The Second Affiliated Hospital and Yuying Children's Hospital of Wenzhou Medical University, Wenzhou, China
[3] The Center for Reproductive Medicine, Department of Obstetrics and Gynecology, Taizhou People's Hospital, Taizhou/The Fifth Affiliated Hospital of Nantong University, Taizhou, China
[4] Department of Clinical Laboratory, The Second Affiliated Hospital and Yuying Children's Hospital of Wenzhou Medical University, Wenzhou, China
* These authors contributed equally to this work.

## ABSTRACT

Autophagy is regarded as an essential process for maintaining cell homeostasis. However, its role in the regulation of sperm motility is still less understood. In this study, we found that mitochondrial oxidative species (mROS) levels were elevated, but ATP levels and mitochondrial membrane potential (MMP) levels were reduced in human sperm with low motility (also called asthenozoospermia, AS) as compared to normal human sperm (normal sperm, NS). Immunocytochemistry staining showed that LC3 was mainly located in the neck of sperm. Western blot analysis showed that AS patients had elevated levels of the autophagy-related proteins LC3, Atg5, Atg7, and Beclin1. Flow cytometry showed that 3-MA treatment reduced sperm motility and MMP, but increased mROS. The results indicate that autophagy is essential for the clearance of mROS in sperm and the maintenance of mitochondrial function.

## INTRODUCTION

Low sperm motility (also called asthenozoospermia, AS) is one of the major causes of infertility in males worldwide (*Baker, 1994*). It has been proven that reactive oxygen species (ROS) play a critical role in regulating normal sperm motility (*Hecht & Zick, 1992*; *Ichikawa et al., 1999*), while ROS overproduction could impair sperm motility (*John Aitken, Clarkson & Fishel, 1989*; *Gomez, Irvine & Aitken, 2002*; *de Lamirande, 1997*). Therefore, eliminating redundant ROS in sperm may be a strategy to improve sperm motility. However, it is still less understood about the mechanisms underlying ROS clearance.

Mitochondria produce the majority of ATP in sperm, but they are also the major source of ROS. A previous study showed that abnormal mitochondria accumulate in the sperm of

Corresponding authors
Hongshan Ge, dafeng7621@126.com
Xinmei Wu,
applewuxinmei@126.com

AS patients, which in turn leads to mitochondrial ROS (mROS) overproduction and a rise in total cellular ROS (*Nowicka-Bauer et al., 2018*). Excessive ROS can interrupt the integrity of sperm membranes, impair sperm motility and eventually induce infertility.

Autophagy plays a crucial role in maintaining cellular homeostasis by removing damaged proteins and organelles (*Mizushima, 2007*) and is also involved in the entire fate of the sperm (*Wang et al., 2022*). For example, atg7, a key protein in initiating autophagy, is involved in acrosome biogenesis, spermatozoa flagella biogenesis and cytoplasm removal during spermiogenesis (*Shang et al., 2016*). Moreover, *Liu et al. (2016)* showed that autophagy dysfunction would perturb cytoskeleton assembly and may ultimately affects spermatids movement and release during spermiation. A previous study demonstrated that autophagy could be activated to degrade damaged proteins induced by ROS in multiple cell types (*Li et al., 2015*). Moreover, the activated autophagy process *per se* could also reduce ROS levels (*Li et al., 2015*). To date, the role of autophagy in the regulation of mROS in patients with asthenozoospermia is still not fully elucidated. In this study, we detected the levels of autophagy-related proteins to investigate the activity of autophagy in asthenozoospermia. Furthermore, we used 3-MA, an autophagy inhibitor, to study the effect of autophagy blockade on normal human sperm. Collectively, this study suggests that autophagy probably plays a protective role in patients with asthenozoospermia partly through clearing mROS.

## MATERIALS AND METHODS

### Sperm preparation

Human sperm samples were collected from 200 donors of proven fertility according to World Health Organization (WHO) criteria. All participants signed an informed consent and the study was approved by The Second Affiliated Hospital Ethics Committee of Wenzhou Medical University (KYKT2018-72). Before we performed this study, strict selection criteria had been made for the enrolled subjects, including men between the ages of 25–35 years old, with normal physical examination (including normal secondary sexual characteristics, penis, spermatic cord, vas deferens and peripheral karyotype). Men with no smoking history or who had quit smoking for at least 3 years were recruited. Semen samples were collected by masturbation after 2–7 days of abstinence. The semen samples were analyzed by computer-aided sperm analysis (CASA, IVOS, Hamilton, OH, USA) and identified as normozoospermia (NS, $n = 120$) and asthenozoospermia (AS, $n = 80$) groups according to WHO Guidelines. Sperm were liquefied for 15–30 min at 37 °C and isolated by density-gradient centrifugation. Briefly, sperm samples were added on the colloidal silica suspension (Percoll; Sigma Aldrich, St. Louis, MO, USA), then centrifuged at 1,300$g$ for 15 min. The pellets were collected and washed three times using HTF medium (Quinn's; SAGE, New York, NY, USA). Finally, the sample is resuspended in HTF medium.

### Immunofluorescence staining

For immunofluorescence staining, sperm were fixed with 4% paraformaldehyde for 15 min, and then cold 95% ethyl alcohol was applied for permeating the sperm

membrane. Sperm was then washed and blocked in 1% fetal bovine serum (FBS) for 2 h at room temperature. After washing in PBS, sperm were incubated with anti-LC3 antibody (1:200; Sigma, St. Louis, MO, USA) and FITC-conjugated secondary antibody (1:300; Abcam, Waltham, MA, USA). Images were taken using fluorescence microscopy (FluoView FV500; Olympus, Tokyo, Japan). Three independent experiments were performed.

## Western blotting

For western blotting, sperm samples were first lysed in SDS buffer (Beyotime, Shanghai, China). The cell lysates were then centrifuged for 10 min at 4 °C at 12,000$g$, and the supernatant was reserved. After dilution with the loading buffer (D-1010; Solarbio, Beijing, China), the supernatant was then denatured for 10 min at 95 °C. The equal amount of protein was added and separated by 12% SDS-PAGE, then transferred to PVDF membranes (Millipore, Burlington, MA, USA). Membranes were incubated with the primary antibodies of LC3 (SAB1305638, 1:100; Sigma, St. Louis, MO, USA), AMPK (#2532, 1:1,000; Cell Signaling Technology, Danvers, MA, USA), mTOR (#2972, 1:1,000; Cell Signaling Technology, Danvers, MA, USA), atg5 (#9980, 1:1,000; Cell Signaling Technology, Danvers, MA, USA), atg7 (#8558, 1:1,000; Cell Signaling Technology, Danvers, MA, USA), beclin1 (#3738, 1:1,000; Cell Signaling Technology, Danvers, MA, USA), p62 (#5114, 1:1,000; Cell Signaling Technology, Danvers, MA, USA) and β-Actin (BM0005, 1:200; Boster, Guangdong, China) at 4 °C overnight. After washing the membranes with PBS three times, the membranes were then incubated with secondary antibodies (#205718, 1:2,000; Abcam, Cambridge, UK) at room temperature for 2 h. Using the ECL detection reagents (32209; Thermo Fisher, Waltham, MA, USA) combining Gel Doc XR+ System to visualize the protein bands. Data were finally analyzed using Image Laboratory software. Three independent experiments were performed.

## Measurement of ATP level

The ATP level of sperm was tested using the ATP Detection Kit according to the manufacturer's instructions (Beyotime, Shanghai, China). Sperm samples were lysed and then centrifuged at 12,000$g$ for 10 min at 4 °C. The supernatant was mixed with 100 μL ATP detection buffer and then added to 96-well black plates. The ATP level (nmol/mg) was analyzed using an automatic microplate reader (Biotek, Winooski, VT, USA). Three independent experiments were performed.

## Measurement of mitochondrial membrane potential

The mitochondrial membrane potential (MMP) was analyzed with the JC-1 kit according to the manufacturer's instructions (3520-43-2; Sigma, St. Louis, MO, USA). After washing the sperm samples three times in PBS, the MMP levels were analyzed by flow cytometry (BD, Franklin Lakes, NJ, USA). Sperm were then incubated for 30 min at 37 °C with JC-1 solution (2.5 mg/mL). Each group involved 10,000 sperm, and MMP intensity was

measured based on the ratio of green/red fluorescence intensity. Three independent experiments were conducted.

### Autophagy inhibitor treatment

Normal human sperms were rinsed with PBS and then treated with 3-MA (Selleckchem, Houston, TX, USA) at 0, 2, 5, and 10 mmol/L concentrations. Every related experiment (sperm motility, Mitosox, and JC-1) involved three independent experiments.

### Measurement of ROS content in mitochondria

The total reactive oxygen species (ROS) level generated by mitochondria in sperm was determined using MitoSOX™Red (Invitrogen, Waltham, MA, USA). First, sperm were incubated with 10 μM MitoSOX™Red at 37 °C for 10 min. After that, sperm were rinsed in PBS three times. Finally, the mROS level was analyzed using flow cytometry. Data were quantified using mean fluorescent intensity (MFI). Three independent experiments were performed.

### Starvation media treatment

Human tubal fluid (HTF) is widely used in assisted reproductive procedures because it can provide sperm with basic materials like electrolytes and glucose. HTF mimics the physiological environment of early human embryos (*Tay et al., 1997*), and provides ideal living conditions for sperm. In order to mimic starvation conditions, the normal sperm was treated with FBS-free HTF (Quinn's; SAGE, New York, USA) medium for 0, 5, 6, and 8 h, respectively. In the control group, HTF medium with FBS was used. Autophagy-related proteins were detected using western blot. Three independent experiments were performed.

### Statistical analysis

Data was analyzed by unpaired $t$ test between NS and AS groups or one-way ANOVA followed by a *post-hoc* comparison test using either LSD (when equal variances were assumed) or Dunnett's T3 (when equal variances were not assumed) method when more than two groups were compared. All of the quantitative data shown represent the mean ± SD, with a statistically significant difference defined as $P < 0.05$. Three independent experiments were performed in each study. Statistical analysis was done using GraphPad Prism.

## RESULTS

### ROS overproduction causes reduced sperm motility

Mitochondria are critical for maintaining normal sperm motility. However, reactive oxidative species (ROS) that are generated by mitochondria could also impair sperm motility. Here, we found that mitochondrial ROS level was significantly increased in the AS group when compared to the NS group (Figs. 1A and 1B), indicating that elevated mitochondrial ROS level might be correlated with decreased sperm motility. The total ATP level was detected in both groups in order to assess the relationship between sperm motility and mitochondria. According to Fig. 1C, the total ATP level in the AS group was lower

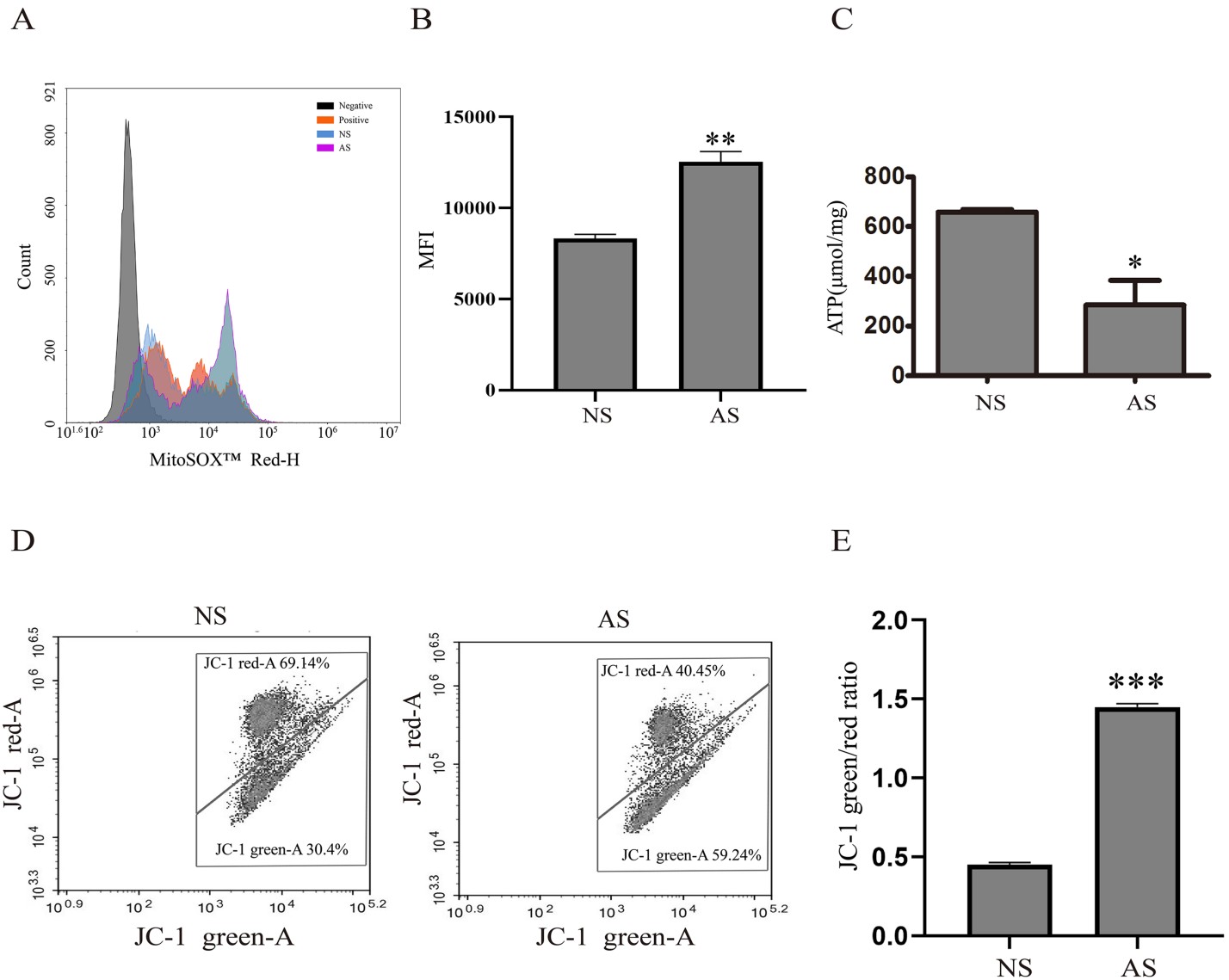

**Figure 1 Reduced sperm motility is related with mitochondrial ROS overproduction.** (A) Sperm collected from the NS and AS group are labeled with MitoSOX for detection of mitochondrial superoxide. Negative, autofluorescrece generated by MitoSOX detection kits. (B) Flow cytometry is used to quantify mean flourecence intensity (MFI). Data represent mean ± SD of three independent experiments. (C) ATP level in the NS and AS group is determained using ATP detection kit. Data represent mean ± SD of three independent experiments. (D) Flow cytometry analysis of the MMP using JC-1 assay in NS and AS group. (E) Quantitative analysis of P4/P3, which represents the JC-1 monomer to JC-1 polymer ratio. Data represent mean ± SD of three independent experiments. NS, normal sperm; AS, asthenozoospermia; MMP, mitochondrial membrane potential. $*P < 0.05$, $**P < 0.005$, $***P < 0.0005$.

than in the NS group, suggesting that mitochondrial metabolism is impaired in sperm with reduced motility. We next performed JC-1 staining to confirm this result. There was a higher green/red ratio in the AS group than in the NS group (Figs. 1D and 1E), suggesting increased mitochondrial depolarization in sperm with reduced motility.

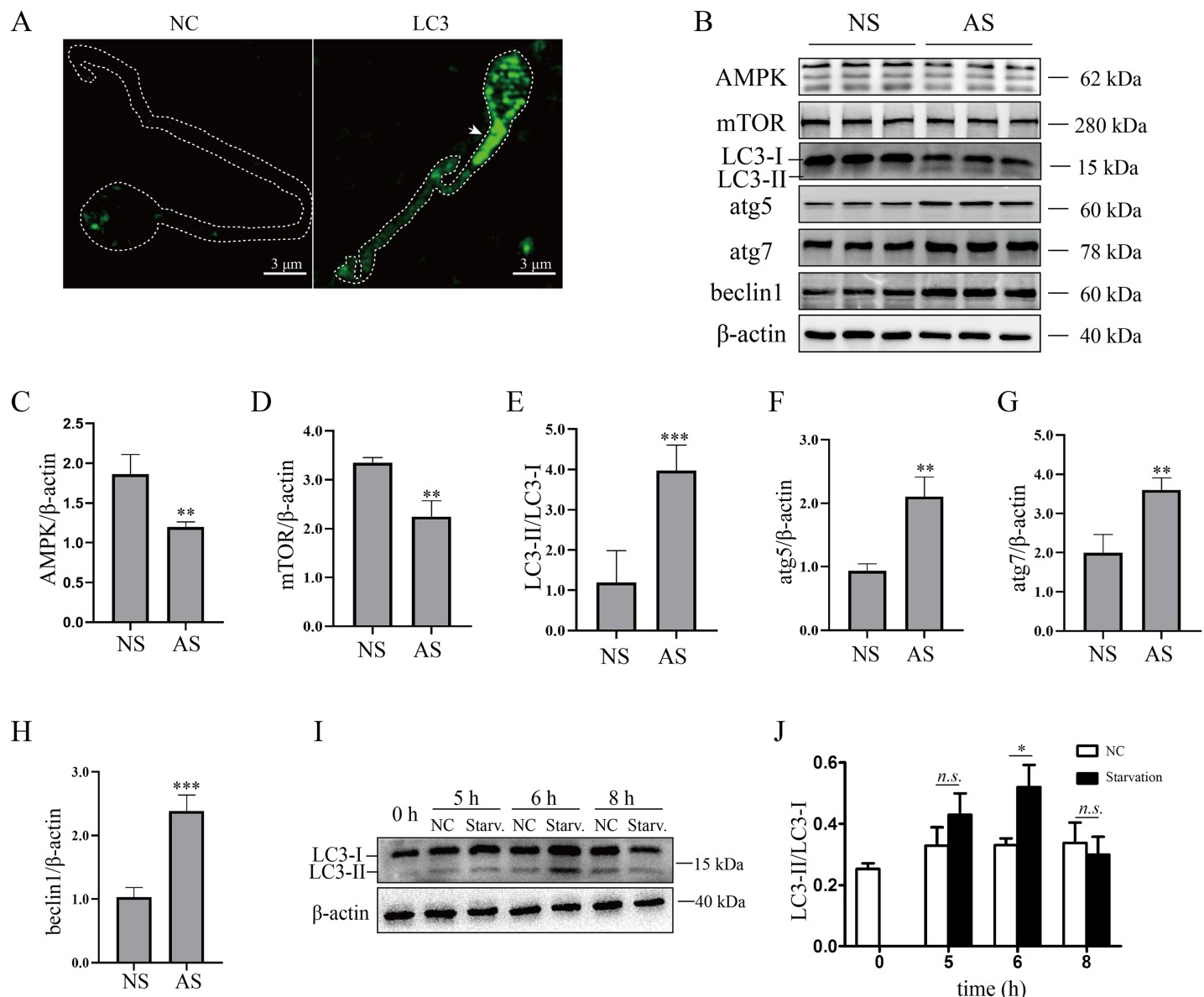

**Figure 2 Autophagy is activated in asthenozoospermia.** (A) Left Panel: negative control. Right Panel: immunofluorescence staining for LC3. The arrowhead indicates the location of LC3-positive area in the sperm. Scale bar, 3 μm; (B) Western blot analysis of AMPK, mTOR, LC3, atg5, atg7, beclin1 and β-actin protein levels in NS (normal saline) and AS (arsenite) treated groups. Molecular weights are indicated on the right. (C) Quantitative analysis of AMPK/β-actin ratio present in (B), three independent tests were performed. (D) Quantitative analysis of mTOR/β-actin ratio present in (B), three independent tests were performed. (E) Quantitative analysis of LC3-II/LC3-I ratio present in (B), three independent tests were performed. (F) Quantitative analysis of atg5/β-actin ratio present in (B), three independent tests were performed. (G) Quantitative analysis of atg7/β-actin ratio present in (B), three independent tests were performed. (H) Quantitative analysis of beclin1/β-actin ratio present in (B), three independent tests were performed. (I) Sperm derived from normal contributors are treated with starvation for 0, 5, 6 and 8 h. Level of LC3-I and LC3-II are shown. (J) Quantitative analysis of LC3-II/LC3-I ratio present in (I), three independent tests were performed. NS, normal sperm; AS, asthenozoospermia. n.s. non-significant, $*P < 0.05$, $**P < 0.005$, $***P < 0.0005$.

## Autophagy is activated in asthenozoospermia

Although mature human sperm may contain an acrosome that functions like a lysosome, there was still disagreement about whether the autophagy process could proceed (*Moreno*

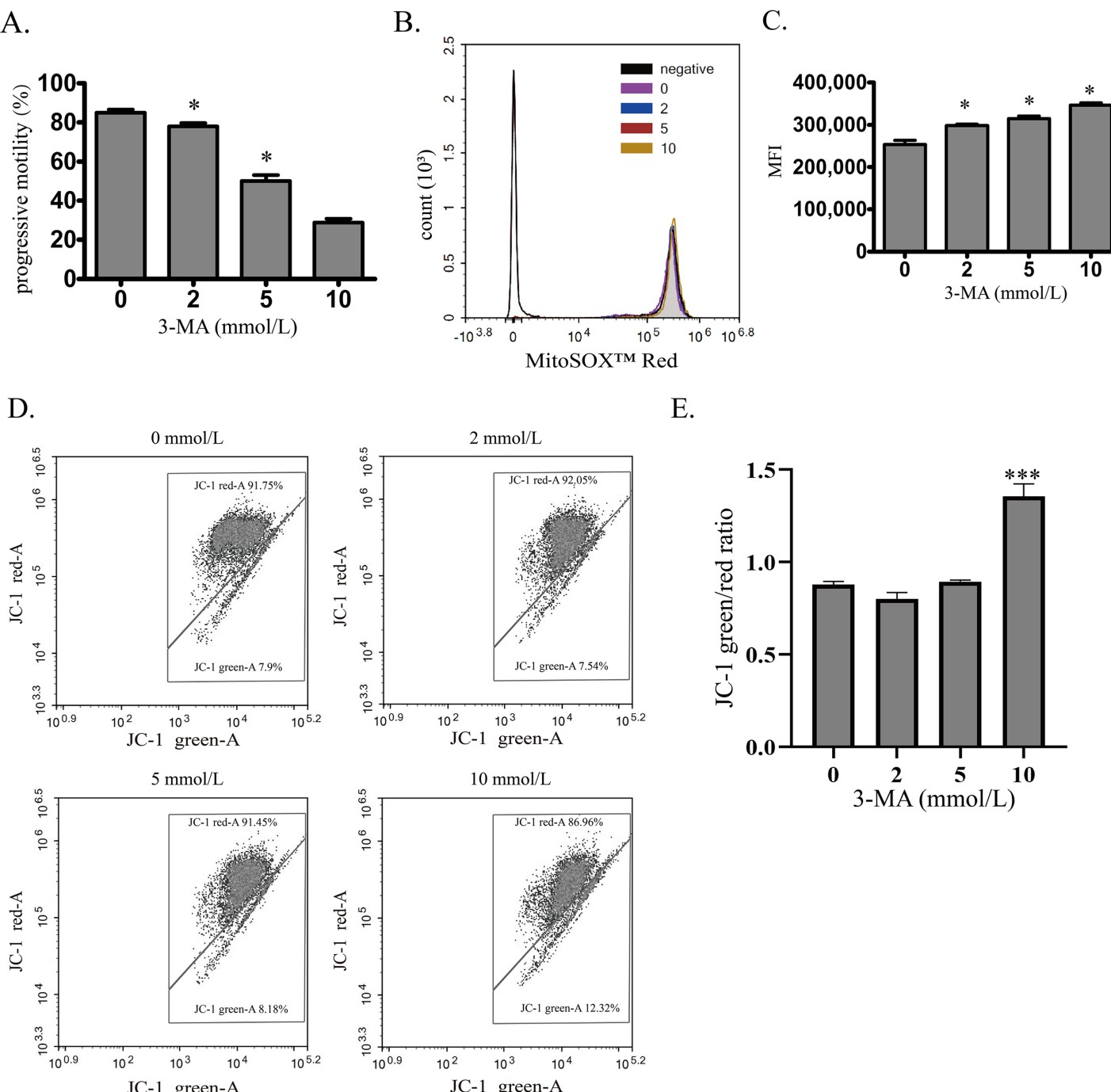

**Figure 3 Autophagy is essential for maintaining sperm motility partly by scavenging mitochondrial ROS and promoting mitochondrial homeostasis.** (A) Progressive motility of sperm after treating with 0, 2, 5 or 10 mmol/L 3-MA is determined. (B) Mitochondrial superoxide level of sperm is detected by MitoSOX probes and quantified after treating with 0, 2, 5 or 10 mmol/L 3-MA. Negative, autofluorescrece generated by MitoSOX detection kits. Three independent tests are performed. (C) Flow cytometry is used to quantify mean fluorescence intensity (MFI). Data represent mean ± SD of three independent experiments. (D) The MMP is determined using JC-1 probes after treating with 0, 2, 5 or 10 mmol/L 3-MA. Three independent tests are performed. (E) Quantitative analysis of P4/P3, which represents the JC-1 monomer to JC-1 polymer ratio. Data represent mean ± SD of three independent experiments. NS, normal sperm; AS, asthenozoospermia; MMP, mitochondrial membrane potential. *$P < 0.05$, ***$P < 0.0005$ (*vs* 0 mmol/L 3-MA).               

& Alvarado, 2006). In the present study, we found that LC3 was enriched in the sperm necks (Fig. 2A), where mitochondria are located. Furthermore, the ratio of LC3-II/LC3-I and the protein levels of atg5, atg7 and beclin1 were elevated in the AS group (Figs. 2B and 2E–2H) compared to the NS group, indicating that autophagy level was increased in human sperm with reduced motility. Then, we detected the proteins that regulate autophagy. As the results showed, protein levels of AMPK and mTOR were decreased in the AS group compared to the NS group (Figs. 2B–2D). We propose that downregulation of mTOR might contribute to the activation of autophagy in the AS group. Furthermore, we found that p62 was increased in the AS group compared to the NS group. Overall, these results indicated that autophagy is upregulated in the AS group. It has been shown that starvation activates autophagy in most mammalian cells. To further demonstrate the effect of starvation on human sperm, human sperm was cultured in carbon starvation. Our results showed that carbon starvation for 6 h increased the ratio of LC3-II/LC3-I (Figs. 2I and 2J). As a result, these data suggest that pathological stimulation in human sperm can induce autophagy.

## Autophagy is essential for maintaining sperm motility partly by scavenging mitochondrial ROS

To investigate the role of autophagy in normal human sperm, autophagy was blocked by 3-MA. With the increased concentration of 3-MA, the sperm motility was decreased in a concentration-dependent manner (Fig. 3A). Moreover, mitochondrial ROS was significantly upregulated by increased 3-MA concentration (Figs. 3B and 3C). These data demonstrate that autophagy may play an important role in maintaining human sperm motility partly through scavenging mitochondrial ROS products in human sperm. We next used JC-1 to estimate the MMP changes after 3-MA treatment (Figs. 3D and 3E). We found that 10 mmol/L 3-MA induced MMP depolarization. In conclusion, these data demonstrate that autophagy plays an important role in maintaining sperm motility partly by scavenging mitochondrial ROS.

## DISCUSSION

In the present study, we investigated the role of autophagy in the regulation of mitochondrial ROS in AS patients. We found that autophagy was increased in the sperm of AS patients. Furthermore, the blockage of autophagy in normal human sperm leads to a reduction in sperm motility. It also led to an increase in mitochondrial ROS levels, and an increase in the number of depolarized mitochondria. These data suggest that autophagy plays a critical role in maintaining normal human sperm motility by removing excessive ROS that are generated by damaged mitochondria.

There is a close correlation between sperm motility and fertility in humans. It is believed that an increased number of sperm with poor motility accounts for the majority of AS cases of infertility as the sperm are unable to reach the oocyte as a result of their poor motility (Chemes & Alvarez Sedo, 2012). Numerous factors have been associated with diminished sperm motility, including ROS accumulation (Gomez, Irvine & Aitken, 2002), mitochondrial dysfunction (Nowicka-Bauer et al., 2018), and excessive DNA

fragmentation (*Barroso, Morshedi & Oehninger, 2000*). Due to their high content of polyunsaturated fatty acids, sperm are highly sensitive to ROS damage (*Aitken & Clarkson, 1987*). Intriguingly, the level of mitochondrial ROS is positively correlated with the levels of ATP and MMP. This is due to the fact that electrochemical gradients generated by proton influx in mitochondria could produce ATP as well as mitochondrial ROS (*Schultz & Chan, 2001*). Coinciding with previous studies, our study demonstrates that the concentration of mitochondrial-derived ROS is increased in AS patients. Moreover, decreased ATP levels and an increased number of abnormal mitochondria are also observed in AS patients. Collectively, these results support the notion that excessive mitochondrial ROS are closely related to poor sperm motility.

Autophagy is involved in regulating the survival of sperm under different stress conditions. *Mancilla et al. (2015)* found that autophagy was activated in nutrient deficiency condition as an adaptive stress response in spermatogonial cells. Moreover, autophagy activation could also protect spermatocyte from apoptosis after radiofrequency exposure (*Liu et al., 2014*) or oxidative stress (*Zhang et al., 2016*). *Aparicio et al. (2016)* reported that autophagy was activated in sperm under different stress conditions, including oxidative stress, UV radiation, and changes in temperature. Those studies demonstrate that autophagy affects various aspects of spermatogenesis. However, the activity of autophagy in AS patients is still not fully elucidated. As presented in the current work, autophagosomes can be observed in normal human sperm. In addition, autophagy can be induced under starvation, suggesting that human sperm possess an intact autophagy-dependent response mechanism. As expected, autophagy is activated in the sperm of AS patients. As described above, we propose that it may be related to the increased number of damaged mitochondria and mitochondrial ROS concentration in the sperm of AS patients. Autophagy probably reacts to scavenge those damaged mitochondria and mitochondrial ROS products to maintain sperm motility and protect sperm from apoptosis.

The physiological level of ROS is conducive to spermatogenesis and maturation (*Kothari et al., 2010*). In spite of this, redundant ROS can cause damage to sperm, including to the membrane of the acrosome (*Li et al., 2018*), DNA (*Lopes et al., 1998*) and the capacitation (*Cai et al., 2019*). To further demonstrate the relationship between autophagy and the mitochondrial ROS level, 3-MA was applied to block autophagy in normal sperm. 3-MA treatment caused decreased sperm motility coupled with increased mitochondrial ROS levels in a concentration-dependent manner. Moreover, flow cytometry showed that the number of depolarized mitochondria was increased after 3-MA treatment, suggesting that the number of damaged mitochondria was elevated. Overall, our work demonstrates that autophagy plays a crucial role in scavenging mitochondrial ROS products and maintaining sperm motility in human sperm.

## ACKNOWLEDGEMENTS

We would like thank all personnel of Department of OB & GYN, the Second Affiliated Hospital & Yuying Children's Hospital of Wenzhou Medical University, for their assistance in this work.

### Funding

This work was funded by the Wenzhou Science and Technology Bureau (Y20240026). The funders had no role in study design, data collection and analysis, decision to publish, or preparation of the manuscript.

### Grant Disclosures

The following grant information was disclosed by the authors:
Wenzhou Science and Technology Bureau: Y20240026.

### Competing Interests

The authors declare that they have no competing interests.

### Author Contributions

- Xiaona Wang conceived and designed the experiments, performed the experiments, analyzed the data, prepared figures and/or tables, authored or reviewed drafts of the article, and approved the final draft.
- Shiyuan Huang conceived and designed the experiments, performed the experiments, analyzed the data, authored or reviewed drafts of the article, and approved the final draft.
- Yu Zhao analyzed the data, prepared figures and/or tables, and approved the final draft.
- Hua Chen conceived and designed the experiments, performed the experiments, analyzed the data, prepared figures and/or tables, and approved the final draft.
- Linzhi Yan performed the experiments, authored or reviewed drafts of the article, and approved the final draft.
- Hongshan Ge conceived and designed the experiments, analyzed the data, authored or reviewed drafts of the article, and approved the final draft.
- Xinmei Wu conceived and designed the experiments, analyzed the data, prepared figures and/or tables, authored or reviewed drafts of the article, and approved the final draft.

### Human Ethics

The following information was supplied relating to ethical approvals (*i.e.*, approving body and any reference numbers):

The Second Affiliated Hospital Ethics Committee of Wenzhou Medical University

### Data Availability

The raw measurements are available in the Supplemental File.

### Supplemental Information

Supplemental information for this article can be found online at http://dx.doi.org/10.7717/peerj.18827#supplemental-information.

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
