# Peer review of "Activation of autophagy is required for clearance of mitochondrial ROS in patients with asthenozoospermia"

_PeerJ, doi:10.7717/peerj.18827_

## Round 0.1 · original submission · Major Revisions

The referees indicate that your paper could be of interest to the field, but that it requires substantial revision, with one of the referees recommending that this version be rejected. Thus, I urge you to carefully consider each of the comments and address them to the fullest extent possible in a revised manuscript, as a paper with only minor revisions is not likely to do well in a subsequent review.

Reviewer 1 ·

Basic reporting

Although the language could use some work, it is minor. The structure and background are appropriate, and the paper is straightforward in terms of what it aims to report. My main criticism is that the organization, conclusions and title are not accurate. The authors do not monitor autophagy, as they state that is not really viable in sperm. What they have is evidence that there are markers of autophagy in sperm (which was known) and that a pharmacological agent that influences autophagy in other cell types may have an effect on sperm function. Given the many well-known unspecific effects of these agents other effects are certainly possible and should have been accounted for.

Experimental design

There are a few issues the authors should address.
- The samples used are very poorly and inadequately characterized, and this is crucial in a study involving human sperm. It is rare to have pure Asthenozoospermic samples (normal count and morphology, low motility), when compared to normozoospermic samples (all parameters normal), and not have oligo or terato samples, or combinations herein. In supplemental information a table should be included of the parameters in both groups to prove that motility is the only difference. Furthermore other parameters that do not directly relate to sperm analysis but are well known to affect sample quality (age, BMI, habits such as smoking, etc.) should be included, as is usual in all papers in this field.
-The authors in general do not include the number of distinct experiments, repeats, numbers of cell analyzed, etc.
-A positive control for Fig 1A should be included.
-Figure 2A seems saturated, rendering the labeling is less convincing as the patterns shown are common in unspecific sperm staining (notably the equatorial segment and tail). A negative control does not seem to have been performed. Also, images from both types should be shown. Some cells seem to have cytoplasmic droplets which could severely influence the WB.
-In Fig2C the differences are also not convincing with starvation, given the variability on the WB and that only one time point shows any differences. Sperm WB are well-known for this sort of variability.
-The authors should show their sperm data controlling for live sperm, cells, which they do not seem to have done. The most simple explanation for the data shown with the autophagy inhibitor 3-MA is that it is actually toxic to sperm, which is the case with many other similar substances. With the data presented it is not possible to determine this. If the inhibitor is prepared in DMSO (or another similar solvent) this also needs to be controlled for.

Validity of the findings

Please see points 1 and 2, which reflect on this. I am not at all convinced that the authors showed what they claim to have shown.

Reviewer 2 ·

Basic reporting

The study presents some results which are not novel in the field, particularly those related to the impact of mitochondrial ROS, ATP levels and mitochondrial membrane potential in association with sperm motility (asthenozoospermia). The novelty of the study relies on the topic of autophagy, since it is an interesting topic related to sperm physiology. However, I see some points that need to be addressed before the article can be published. Overall, it is a simple article, not very well written, and English needs to be improved. Also, importantly, only LC3 proteins were analyzed to evaluate autophagy activation, would be desirable to analyze other autophagy-related proteins as well. The experimental design do not completely support the author`s claims, tI here are some more experiments needed to support the interpretation made by the authors. Also, the flow cytometry analyses of JC-1 are not properly compensated, which make not possible to correctly interpret the results. Figures and legend need to be improved, given more details and facilitating the association with the text and legends.

Experimental design

I see some points in the experimental design that need to be addressed.
- The authors analyzed LC3 proteins in AS and NS samples in order to assess autophagy. This is interesting since it suggests autophagy activity in sperm of low quality. However, it would add strength to the manuscript if the authors also analyze other autophagy-related proteins (see article by Aparicio et al., 2016), in order to present a more detailed view of autophagy activation in asthenozoospermic men.
- Some more experiments are necessary to properly interpret the results of autophagy blicking by 3-MA. L134 – 136: It is necessary to evaluate sperm viability under these experimental conditions, in order to confirm that the concentrations used are not toxic to human spermatozoa. This will allow to suggest, in a more accurate manner, that the decrease in sperm motility is due to the autophagy blocking and not due to a toxicity of the compound itself.
- The authors used some technologies that are strong to analyze sperm parameters (flow cytometry, WB, ATP levels, however, some flow cytometry analysis needs to be improved. Also, some methodology details are lacking, which avoid to properly interpret the results. Detailed suggestions are made in "Additional comments".

Validity of the findings

In my opinion some extra experiments are necessary to support the claims of the authors. In the current state of the manuscript I see that the only novel results the manuscript present is the increase in LC3-II in AS patients. The results presented in figure 1 are not novel since previous report have already demonstrated these results. Results in figure 3 need more experiments to support the conclusions given. Detailed suggestions are made in "Additional comments".

Additional comments

General comments:
- Title: The title do not reflect the results presented in the manuscript.
- The abstract needs to be improved. I suggest to include the main objective of the study as well as a brief description of the experimental design and methodology.
- Some sentences in the introduction are not properly cited, I suggest to include all references.
- The introduction needs improvements, including detailed information about the current knowledge regarding autophagy in sperm cells.
- Part of the introduction includes the results and conclusion of the study (L42 – 48), I suggest to remove this paragraph and replace it by the knowledge gap and the main objective of the study.
- The English needs to be improved.
- L65 –L67, L75: would be useful to indicate the catalog number of the antibody used.
- L 62 – 68: I suggest to rewrite this section in a more appropriate scientific grammar.

Material and methods:
- L 87 – 88: Please state clearly the final concentration of JC-1, perhaps by rewriting the paragraph.
- Material and methods: L 98 – 100: Please, include more details about the FBS-free HTF, which was the rationale of using this medium?
- Material and methods: L 98 – 100: It is necessary to include the number of experiments performed here and the characteristics of the sperm donors and samples used in this experiment.
- Statistical analysis: would be desirable to present the results as mean  SD (instead of s.e.m.) in order to proper visualize the real dispersion of data.
- Statistical analysis: it is not stated which test did the authors use to compare the AS and NS groups.
- The experiments related to the autophagy blockage are not described in material and methods. It is necessary to include the compound used, the number of experiments performed, the characteristics of donors and samples used.

Results:
- Figure 1: the legend is not clear nor detailed enough.
- The data showed in figure 1 are not novel, it is widely recognized the important role played by mitochondrial ROS as a cause of decreased sperm motility in asthenozoospermic patients (e.g.: DOI: 10.26402/jpp.2018.3.05). Similarly, it is widely recognized the association between decreased ATP levels and MMP with decreased sperm motility (asthenozoospermia).
- Figure 1A: It is not stated what you considered “negative”. It that the autofluorescence? It should be properly indicated.
- Figure 1C: The dot plots depicting the data of flow cytometry analysis of JC-1 show that the green and red fluorescence are not properly compensated. This makes not possible to properly interpret the presented data. This is a very important point. I suggest the authors to contact an expert in flow cytometry who is able to properly compensate the green and red fluorescence and then to analyze the data again.
- Figure 1C (graph): are the data really depicted in percentage? I am not sure that this is ok, since you are presenting the ratio of green/red fluorescence.
- Figure 2: the letters assigned to the images do not make easy the reading of the figure, there are some graphs lacking a letter to guide the association with the text in the manuscript or the figure legend (which is also not detailed enough).
- Figure 2 B and C: Would be desirable that the authors present the entire WB image, including all MW markers (not only the 15 and 45 kDa), in order to visualize the specificity of the antibody used. Perhaps as a supplementary representative image.
- Results L121 – 123: Please provide proper references to this sentence.
- Results: L129 – 130: the authors claim that “Our result showed that LC3-II/LC3-I ratio was increased in a time-dependent manner”, however, the results do not show that, since at 8 h no increase in the ratio was observed. Please, clarify this.
- Results L 136 – 137 and Figure 3B: The histogram presenting the MitoSOX red fluorescence profiles do not evidence differences among the concentrations used.
- Figure legend 3B is not detailed enough. Information about the number of experiments is necessary.
- Figure 3. Some images in the figure lack a proper letter that would allow to easily read the figure in association with the text and legend.
- Figure 3. Again, the analysis of JC-1 is not properly compensated, which make not possible to interpret this result.
- I suggest the authors to perform WB to analyze the LC3 proteins in the conditions of autophagy blockade by 3-MA, in order to demonstrate that under these experimental conditions, autophagy is in fact blocked. This would add more information to the manuscript, would strength the claims of the authors as well as the interpretation of data.

Discussion. It includes basic analysis of data presented, supported by proper references. It should be improved considering the suggestions made before.

---

## Round 0.2 · Minor Revisions

I have taken over handling this manuscript, and apologise for the delay. A few minor issues remain.

Please can you clearly state on each figure legend how many biological and technical replicates were performed for each experiment (i.e. FACS, blots, etc.). Also make sure that the primary data for ALL biological replicates is uploaded, not just the uncropped blots shown in the manuscript. You need to show all the data for all the biological replicates.

Once these minor issues are sorted, I can accept the paper.

Reviewer 2 ·

Basic reporting

The authors have addressed most of the comments and suggestions made for this reviewer. I consider that in its current status the manuscript is suitable for publication.

Experimental design

no comment

Validity of the findings

no comment

Additional comments

no comment

---

## Round 0.3 · accepted · Accept

Thank you for attending to these final few comments. I am happy to recommend acceptance now.